# Buckwheat Disease Recognition Based on Convolution Neural Network

**Xiaojuan Liu [1], Shangbo Zhou [1,*] , Shanxiong Chen [2], Zelin Yi [3], Hongyu Pan [2] and Rui Yao [2]**

1   College of Computer Science, Chongqing University, Chongqing 400044, China; 20181401008@cqu.edu.cn
2   College of Computer and Information Science, Southwest University, Chongqing 400715, China;
    csxpml@163.com (S.C.); phy2020hongyu@163.com (H.P.); 13234029853@163.com (R.Y.)
3   College of Agronomy and Biotechnology, Southwest University, Chongqing 400715, China;
    yizelin0326@163.com
*   Correspondence: shbzhou@cqu.edu.cn

**Abstract:** Buckwheat is an important cereal crop with high nutritional and health value. Buckwheat disease greatly affects the quality and yield of buckwheat. The real-time monitoring of disease is an essential part of ensuring the development of the buckwheat industry. In this research work, we proposed an automated way to identify buckwheat diseases. It was achieved by integrating a convolutional neural network (CNN) with the image processing technology. Firstly, the proposed approach would detect the buckwheat disease area accurately. Then, to improve the accuracy of classification, a two-level inception structure was added to the traditional convolutional neural network for accurate feature extraction. It also helps to handle low-quality image problems, which includes complex imaging environment and leaf crossing in sampling buckwheat image, etc. At the same time, instead of the traditional convolution, the convolution based on cosine similarity was adopted to reduce the influence of uneven illumination during the imaging. The experiment proved that the revised convolution enabled better feature extraction within samples with uneven illumination. Finally, the experiment results showed that the accuracy, recall, and F1-measure of the disease detection reached 97.54, 96.38, and 97.82%, respectively. For identifying disease categories, the mean values of precision, recall, and F1-measure were 84.86, 85.78, and 85.4%. Our method has provided important technical support for realizing the automatic recognition of buckwheat diseases.

**Keywords:** buckwheat disease; convolutional neural network; image detection; deep learning; recognition

## 1. Introduction

### 1.1. The Significance of Buckwheat Disease Identification

Buckwheat is an important grain with abundant nutrition, containing protein, cellulose, sugar, and antioxidant rutin that are very beneficial to human health. Moreover, buckwheat is a high-quality crop with development potential due to its strong planting adaptability, cold tolerance, and poor soil adaptability. In China, Buckwheat is mainly distributed in the high altitude and cold mountainous areas, including the northwest, the northeast, and southwest. Buckwheat is the main food and economic crop in these areas [1]. Globally, buckwheat is mainly distributed in Canada, India, Japan, and other countries [2].

However, buckwheat disease affects the yield and quality of buckwheat greatly and degrades nutrition and quality. The buckwheat disease is one of the most critical agricultural natural disasters in the world. There are many types of buckwheat disease, which have a great impact and often cause disasters [3]. In recent years, the buckwheat planting area has been gradually expanded. Buckwheat planting has changed from traditional small-scale cultivation to modern mechanized cultivation, and therefore, the requirements for disease control are also increasing. Diagnosing buckwheat disease accurately and in a timely manner is an important means of prevention and control of the disease [4].

### 1.2. Disease Recognition Based on Deep Learning in Agriculture

Conventional disease recognition has been limited by slow speed, strong subjectivity, high misjudgment rate, and inefficiency, therefore, it no longer meets the needs of modern agricultural production. Buckwheat mostly grows in mountainous areas, so there is a shortage of agricultural technicians, often leading to missed optimal periods of disease control [5]. Fortunately, with the development of machine learning and pattern recognition, it is feasible to classify and identify buckwheat pests and diseases in agriculture. In recent years, many new methods of pattern recognition have emerged for image classification, detection, and recognition. These technologies have been beneficial in improving the identification efficiency, reducing the cost, increasing the identification accuracy, and easing the burden of experts [6–8]. Researchers have developed the classifiers of pests and diseases in agricultural crops by support vector machine [9], K-means clustering [10], radial basis function [11], genetic algorithm [12], Bayesian classification [13], integrated learning [14], filter segmentation [15], and so on, and have achieved good results. With the rise of deep learning, deep structure learning technologies, such as convolutional neural networks and recurrent neural networks have made constant progress. Much attention has been paid to carrying out the automatic identification of pests and diseases by using deep learning techniques.

In the above research, the recognition of crop diseases mainly focuses on the field crops. Field crops are planted extensively and it is relatively easy to collect and select samples; the sample database established is standard. In the image samples, morbid and healthy crops are relatively clear and the image quality is good; therefore, the current deep learning framework can achieve a better recognition effect on these field crops. However, as a multigrain crop, buckwheat has no standard image database and is mostly planted in mountainous areas; there exist many limitations for sampling, including that the illumination of samples is uneven, and the leaves are overlapping, etc. As a consequence, the image of buckwheat is low quality, containing serious noise.

In this paper, a two-level inception structure is inserted into the basic framework of the convolutional neural network to accurately extract the features of buckwheat images and improve the accuracy of classification as well. The proposed approach is capable of dealing with overlapping leaves and low-quality images. Meanwhile, in order to reduce the influence of light in the sampling process, convolution based on cosine similarity is adopted instead of traditional convolution operation, so that better feature extraction can be carried out for samples with uneven light, and finally, accurate recognition of buckwheat disease can be realized.

In agricultural production, automatic classification has been widely applied to crop disease images, which is a key technology for pesticide selection and spraying in precision agriculture [16]. Feature extraction is an important technique for identifying crop diseases. However, it is not flexible to traditional classification algorithms based on artificial feature extraction, as it requires high requirements of professional knowledge, high time complexity, and difficulty in extracting high-quality features [17]. Deep learning can obtain the multi-scale feature of crop diseases, realize the characteristic expression of different diseases more accurately, and it is beneficial to the accurate identification of crop diseases [18]. Tongke Fan et al. presented a deblurring method for local blurred images based on deep learning. By training convolution neural network models with different structures, a normalized segmentation algorithm based on spectral theory was used to segment the image of plant diseases and insect pests. This method has good robustness, generalization, and accuracy for the segmentation of pests and diseases in agricultural disease image recognition [19]. Qiang Dai et al. proposed a generative adversarial network with dual-attention and topology-fusion mechanisms (DATFGAN), which can effectively transform unclear images into clear and high-resolution images. The weight sharing scheme in DATFGAN can significantly reduce the number of parameters. For recognizing the processed image, the result shows that DATFGAN is superior to other methods and has sufficient robustness [20].

In the field of deep learning, many learning frameworks have been developed for image feature extraction and applied in agricultural pest and disease detection. Rahman et al. proposed a detection method for rice pests based on deep learning [21]. They used VGG16 and Inception V3 structures and fine-tuned them to detect and identify rice pests and diseases. Chen, Junde et al. used a migration learning mechanism to obtain the DenseNet network through pre-training on the ImageNet database and Inception module, which is used for the identification of rice pests and diseases. Compared with other methods, this method has better recognition performance and lower training costs [22]. Li Dengshan et al. proposed a video detection architecture based on deep learning to detect plant diseases and insect pests by video [23]. In addition, this paper proposes a set of video detection evaluation indexes based on machine learning classifier, which can effectively evaluate the quality of video detection. Compared with VGG16, ResNet-50, ResNet-101, and YOLOv3, this network structure system is more suitable for detecting diseases and pests in untrained rice videos. In the literature [24], CNN is also used to identify maize leaf disease. It can identify three major maize diseases in southern Africa; maize leaf blight, common rust (sorghum rust), and gray spot (brown spot). Krishnaswamy et al. used VGG16 as the eighth convolution layer feature extractor and multi-class support vector machine (MSVM) to classify diseases and pests in eggplant, and achieved good results [25]. On the basis of analyzing state-of-the-art convolution neural networks, AlexNet, GoogleNet, Inception V3, ResNet18, and ResNet 50, Valeria et al. used the improved GoogleNet model to identify tomato pests and diseases [26].

Furthermore, Fuentes et al. combined visual object recognition and language generation models to generate detailed information about plant abnormal symptoms and scene interactions. In the identification task of tomato pests and disease, the method achieved 92.5% accuracy in the dataset of tomato plant anomaly [27]. Xiao Qingxin et al. used the Aprioro algorithm to find the association rules between climatic factors and the occurrence of cotton pests and diseases, the prediction of pests and diseases was attributed to time series prediction, and a prediction method based on LSTM (long short-term memory) was presented. The results show that suitable temperature, humidity, low rainfall, low wind speed, suitable sunshine time, and low evaporation are the main factors causing cotton pests and diseases [28]. Verma, S. et al. used Capsule Net to classify potato diseases and achieved good results [29].

At present, the detection of crop diseases is mostly carried out in the standard crop sample database. The standard sample database often has obvious disease characteristics through expert screening, and the imaging conditions and image quality are reliable. The existing deep learning methods for identifying crop diseases and insect pests do not consider the complexity of disease images collected in the field, such as leaf overlap, uneven lighting, shadow coverage, etc. These challenges greatly affect the recognition effect. Therefore, how to adapt to a complex imaging environment and accurately extract key features is an important direction of crop disease recognition research.

Therefore, the form of buckwheat plants, especially the leaf shape, affects the light distribution of the population canopy. Therefore, canopy leaves tend to block the middle and lower leaves, and in the process of image collection, the shaded areas formed by occlusion often causes great interference to the discrimination of diseases. In this paper, the recognition of buckwheat disease includes two processes: detection and recognition of disease area. First, the MSER (Maximally Stable Extremal Regions) method is used to detect disease area, and further, the improved AlexNet network accurately implements disease area detection. Then the inception structure and cosine similarity convolution are used to complete the discrimination of specific diseases.

The rest of the paper is organized as follows: the network structure of CNN are discussed in Section 2; materials and data processing is introduced in Section 3; the method of Buckwheat disease recognition is presented; the achieved experimental results and related discussions are presented in Section 4; finally, Section 5 draws up our conclusions of the study and future directions.

## 2. Network Structure of CNN

CNN is a feed forward neural network including convolution operation and with a deep structure. Its basic structure includes input layer, conv layer, pooling layer, full connection layer and output layer (classifier) [30]. A typical CNN structure is shown in Figure 1.

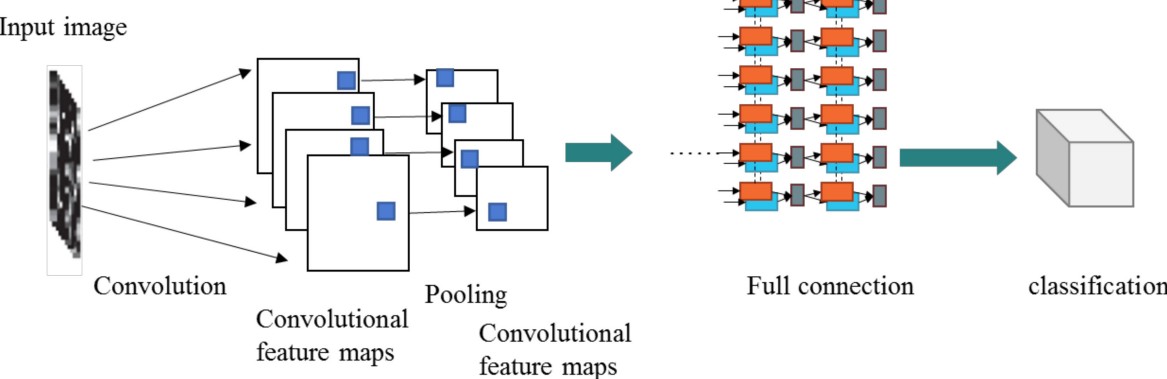

**Figure 1.** Typical convolution neural network structure.

In the convolution network, the transformation between layers is a process of feature extraction. Each layer is composed of several two-dimensional planes, each plane represents a feature map (FM). The input layer is the original image, and each feature extraction layer (convolution layer) in the network is followed by a secondary extraction calculation layer (pooling layer). This structure of secondary feature extraction enables the convolutional network to have a certain tolerance when there is a large deformation of the input data. Generally, there are several convolution layers and pooling layers. The specific operation process is as follows:

(1) Convolution process: The input image (or FM of the upper layer) is convoluted with a trainable filter and then the bias $b_x$ is added. The convolution layer $C_x$ is obtained.

(2) Pooling process: Pixels in the neighborhood calculate the average value to get a pixel, which is weighted by scalar $W_{x+1}$, then offset $b_{x+1}$ is added to the weighted results. Using an activation function, a reduced feature map $S_{x+1}$ can be obtained.

(3) The full connection layer: The full connection layer is equivalent to the hidden layer in multilayer perceptron (MLP). It is fully connected to the previous layer. The calculation process is to multiply the output result of the upper layer by the weight vector, add a bias, and then pass it to the sigmoid function.

(4) The output layer (classification layer): It consists of Euclidean radial basis function units, and each category corresponds to a unit. The output layer uses a classifier to calculate the probability that the input sample belongs to a category.

## 3. Materials and Data Processing

We have established a disease database for buckwheat. The disease images were taken in rural farmland by a Canon EOS 90D digital camera from 2017 to 2019. Buckwheat diseases mainly occur in the leaves. Their occurrence depends on many factors, such as temperature, humidity, rainfall, variety, season, nutrition, and so on. The Chongqing buckwheat industry innovation team of China has carried out extensive research activities in Chongqing and Sichuan Province. In Chongqing, the sampling areas are mainly located in Weituo farm of Hechuan district, Xiema farm of Beibei District, Banxi farm of Youyang District, Fengan farm of Wanzhou District, and Zhongyi farm in Shizhu District; in Sichuan the sampling areas are mainly located in Liangshan, involving Jinqu Township in Zhaojue District, Shaluo Township in Butuo District, Lami Township in Leibo District, and Nanwa Township in Jinyang District. From February 2017 to November 2019, 7230 buckwheat disease images were collected from buckwheat fields. The images with different backgrounds

were collected in real scenes. In order to obtain a group of representative images as much as possible, the images were collected in spring and autumn, and the weather conditions included "less cloud", "cloudy", and "overcast". These steps improve the robustness of the model.

The database contains eight types of disease images, buckwheat spot blight, buckwheat sclerotinia, buckwheat seedling blight, buckwheat ring spot, buckwheat downy mildew, buckwheat brown spot, buckwheat virus disease, and buckwheat white mold. Sample images of each class have been depicted in Figure 2a–h. 5000 images have been selected in this work, which includes 500 images for each disease and 1000 images without disease. The training data set consists of 400 images for each disease and 800 disease-free images. The remaining were used as the test set. A 5-fold cross-validation was used for the evaluation.

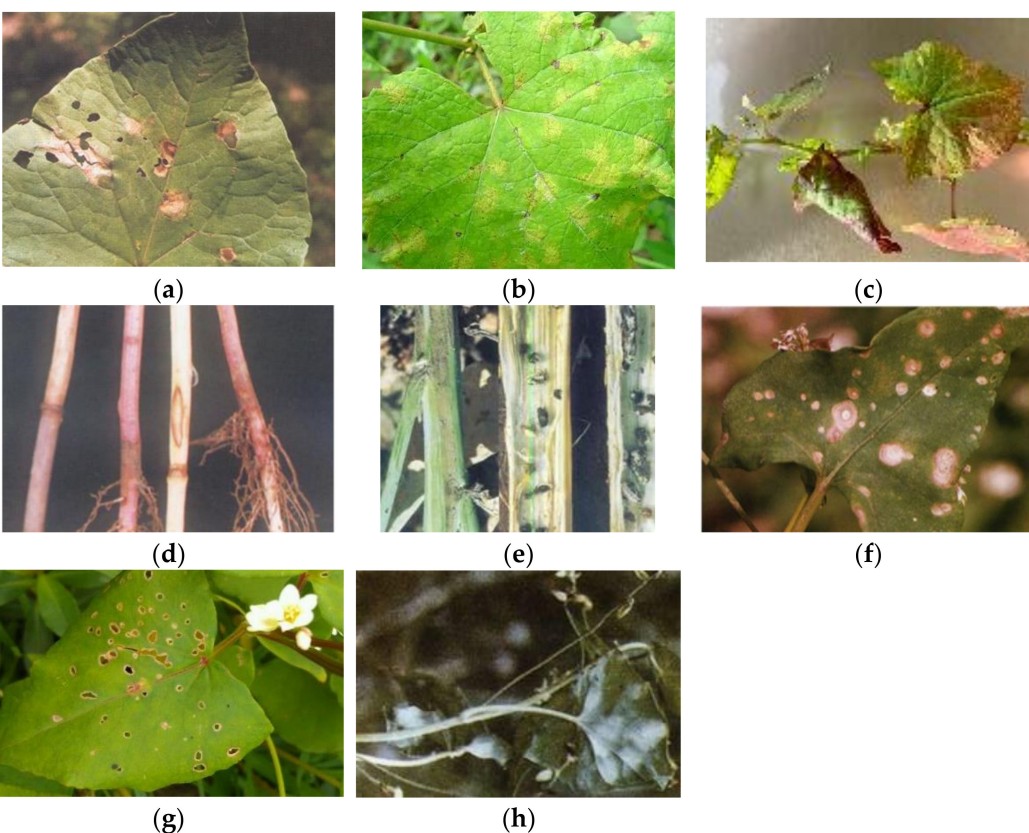

**Figure 2.** A sample image of each detected class. (**a**) Spot blight; (**b**) Downy mildew; (**c**) Virus disease; (**d**) Seedling blight; (**e**) Sclerotinia; (**f**) Ring spot; (**g**) Brown spot; (**h**) White mold.

## 4. Buckwheat Disease Recognition Based on Convolution Neural Network

### 4.1. Detection of Disease Area

Buckwheat pests and diseases often occur on the leaves [31]. Usually, the whole leaf is used as the input to train the recognition model of crop diseases to get the best classification structure. This method originates from the early application of deep learning in image classification. It takes advantage of the simple structure of the model constructed and the high efficiency of training and recognition. However, the whole image contains a lot of feature information, many of which are not highly relevant to the recognition task and interferes with model performance. There are many kinds of buckwheat diseases, and some diseases are similar in leaves, which is easy to produce misidentification. For example, spot blight and brown spot are easily confusing. If we can locate the disease area of buckwheat, make the recognition model only focus on the disease area, and accurately extract the most important feature of buckwheat disease, it will improve the recognition accuracy of different types of disease. Our task is to identify the categories of buckwheat diseases. It is necessary to accurately detect the buckwheat disease occurrence region from the image

so as to extract the features of the disease region. Based on the regional feature extracted, different types of diseases can be effectively identified to ensure classification accuracy. The region detection for buckwheat disease is to separate the disease region and non-disease region from the images, and then send the disease region into the network to complete the training and recognition. For the detection of buckwheat disease area, this paper proposed a method combining (MSER) [32] with CNN to detect the buckwheat disease regions. The detailed steps are as follows:

　　　Step 1: Using MSER to detect disease areas of buckwheat, the MSER algorithm is as follows:

(1)　The image of buckwheat disease was grayed, and the gray image was binarized with 256 different thresholds in the gray range (0–255); Let $Q_t$ denote a connected region in the binary image corresponding to the binarization threshold $t$. When the threshold of binarization changes from $t$ to $t + \Delta$ and $t - \Delta$, $\Delta$ is the change value, the connected region $Q_t$ becomes $Q_{t + \Delta}$ and $Q_{t-\Delta}$ correspondingly.

(2)　Calculating the area ratio when the threshold is $t$, $q(t) = |Q_{t+\Delta} - Q_{t-\Delta}|/|Q_t|$. When the area of $Q_t$ changes slightly with the change of the binarization threshold $t$, namely, $q_t$ is the local minimum, $Q_t$ is the Stable extremum region. Where $|Q_t|$ is the area about connected region $Q_t$. $|Q_{t+\Delta} - Q_{t-\Delta}|$ is the area of the remaining region after $Q_{t+\Delta}$ subtracts $Q_{t-\Delta}$

　　In the process of MSER detection, some large rectangular boxes could contain small ones, so these regions should be merged and the small rectangular boxes should be removed. For the merging of two regions, let the parameters of connected region 1 are $\beta_1, \chi_1, \delta_1, \varepsilon_1$ and the parameters of connected region 2 are $\beta_2, \chi_2, \delta_2, \varepsilon_2$ where $\chi$ and $\beta$ represent the minimum and maximum values of the minimum circumscribed rectangle in the *y*-axis direction of the connected region. $\delta$ and $\varepsilon$ represent the minimum and maximum values of the minimum circumscribed rectangle in the *x*-axis direction of the connected region, then the connected region 1 contains the connected region 2, which can be determined according to Equation (1):

$$\begin{cases} \chi_1 \leq \chi_2 \\ \beta_1 \geq \beta_2 \\ \delta_1 \leq \delta_2 \\ \varepsilon_1 \geq \varepsilon_2 \end{cases} \tag{1}$$

　　Through the above steps, the disease area is selected. However, it can be seen from Figures 3 and 4 that there is still overlap and error detection between disease area and non-disease area. The fuzzy area in the background and blade edge area were incorrectly detected as the disease area.

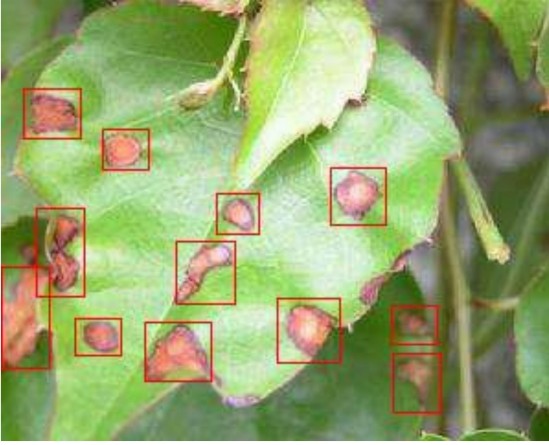

**Figure 3.** Detection effect of buckwheat brown spot disease.

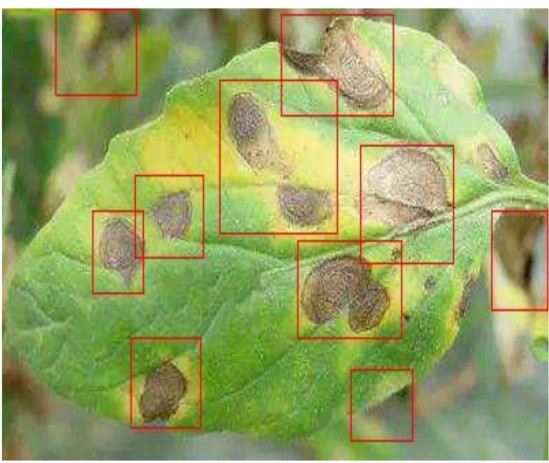

**Figure 4.** Detection effect of buckwheat ring rot disease.

Step 2: For further distinguishing between disease area and non-disease areas, avoiding detection boxes overlap and false detection, we designed a CNN binary classifier based on AlexNet [33]. Its structure is shown in Figure 5. The network has two convolution layers and two pooling layers. The final full connection layer was a binary classifier for disease area and non-disease area. First of all, input a $128 \times 128$ image, and then $16\ 3 \times 3$ convolution kernels are used to extract the features of the input image. Then, a $32 \times 32 \times 16$ convolution layer is obtained. Next, the data dimension of the convolution layer is reduced by using a $2 \times 2$ maximum pooling layer so that we can obtain a $16 \times 16 \times 16$ pixel pooling layer. $32\ 5 \times 5$ convolution kernels are used to further extract higher-level features. Finally, the output of $8 \times 8 \times 32$ is obtained by using a $2 \times 2$ maximum pooling. All the output features are connected in a fully connected layer. The weights of the output features is calculated according to the feature vector. The probability of belonging to two categories is output, so we can locate disease region of the input image.

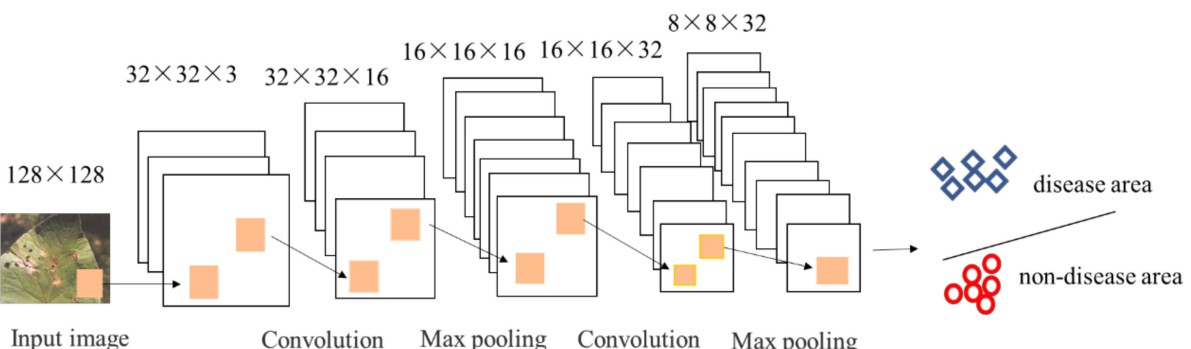

**Figure 5.** The network structure of disease area detection.

### 4.2. Convolutional Neural Network Structure of Buckwheat Disease Recognition

The appropriate framework of CNN is the key to identifying buckwheat diseases. Generally, the performance improvement of convolutional neural networks depends on increasing the depth and width of the network, which is, increasing the number of hidden layers and neurons in each layer. This will lead to a wider parameter range, easier overfitting, and more computing resources. The deeper the network, the easier the gradient will disappear, and it makes optimization difficult. To deal with these challenges, the full connection is changed into a sparse connection, and the convolutional layer also adopts a sparse connection. However, it is inefficient to calculate the asymmetric sparsity, we need to find the optimal local sparse structure, which can be approximated by the convolution network. The inception structure was introduced to tackle this issue.

### 4.2.1. Improved Network Based on Inception Structure

Inception is a local topology network, which performs multiple convolution operations or pooling operations for the input image in parallel, and stitches all the output into a very deep feature map. Because different convolution operations and pooling operations such as $1 \times 1$, $3 \times 3$ or $5 \times 5$ can obtain different information of buckwheat disease image, parallel processing of these operations and combining all the results will obtain better image characterization of buckwheat disease.

The convolutional neural network used in this article is shown in Figure 6. The network adds two inceptions based on the traditional structure. The specific processing was as follows:

(1)　Firstly, the input of the network was the image of buckwheat disease with the size of $64 \times 64$. The diseased image was then convolved with 136 $9 \times 9$ convolution kernels to obtain 136 $56 \times 56$ feature maps.

(2)　The feature map was sent to the inception 1 structure, and all inception structures in the network used the same convolution operation, that is, the size of the feature map was not changed. The size of the pooling window in all subsequent pooling layers was $2 \times 2$. Therefore, the size of the feature map becomes $28 \times 28$ after pooling.

(3)　200 $5 \times 5$ convolution kernels are adopted to obtain 200 $24 \times 24$ feature maps. After these feature maps were sent to the inception 2 structure, 264 $5 \times 5$ convolution kernels were used to obtain 264 $8 \times 8$ feature maps after pooling.

(4)　After pooling, it entered the last layer of the convolution layer. The number of convolution kernels was 520 and the size was $3 \times 3$. Therefore, 520 feature maps of $2 \times 2$ will be obtained. Finally, the results were processed by the full connection layer and the classified output layer.

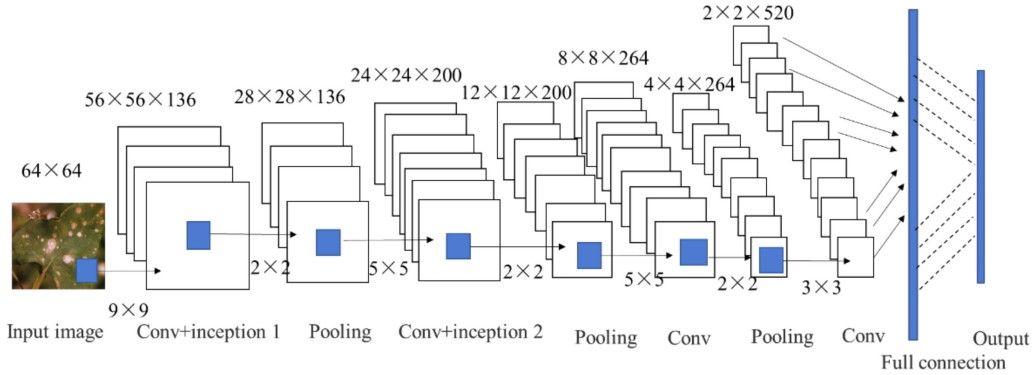

**Figure 6.** The structure of the convolutional neural network for buckwheat disease identification.

In the initial two convolutional layers of the network, the receptive fields of size $9 \times 9$ and $5 \times 5$ were used, respectively. In order to extract more feature information of different scales in a smaller receptive field, and expand the width and depth of the network, we added inception 1 and inception 2 structures to the two convolution layers, respectively. Their structures are shown in Figures 7 and 8. In the inception 1 structure, $1 \times 1$, $3 \times 3$, $1 \times 5$, $5 \times 1$, 4 different scales of convolution kernels are used for multi-channel feature extraction, and the channels are fused. The top $1 \times 1$ convolution can effectively reduce the number of channels in the input feature map and the computation cost of the network. The $1 \times 1$ convolution at the bottom layer is to restore the number of channels in the input feature map and maintain the consistency of the number of channels in the input and output feature maps. In the inception 2 structure, the input feature map becomes smaller, so the $1 \times 5$ and $5 \times 1$ convolutions in inception 1 are replaced with $1 \times 3$ and $3 \times 1$ convolutions, respectively. In the second convolutional layer, the number of input feature maps is more than that of the first layer, so the number of channels in the structure is increased from 30 to 40.

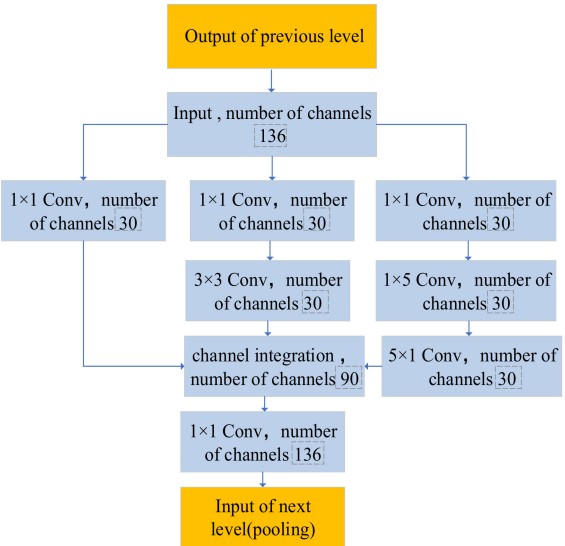

**Figure 7.** Inception 1 structure.

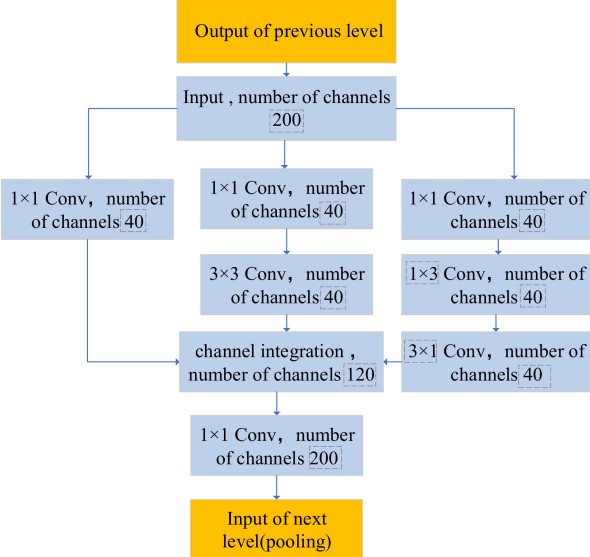

**Figure 8.** Inception 2 structure.

Conv represents the convolution operation according to the size of the convolution kernel; the number of channels represents the number of convolution channels.

### 4.2.2. Convolution Based on Cosine Similarity

Buckwheat disease data are collected from field, which is limited by the sampling environment and disturbed by noise. Therefore, in order to achieve a higher activation value after convolution operation, only the positions with similar characteristics to the convolution kernel can be obtained in the feature diagram. At the same time, it is necessary to reduce the difference between features and avoid the interference of sample noise on feature extraction. In this paper, the idea of cosine similarity was introduced into the operation of the convolution layer [34], and the input feature map and the convolution kernel were regarded as two vectors, and the correlation between them was calculated.

In the traditional convolutional neural network, the output of *J*-th feature map belonging to *l*-th convolution layer can be expressed as follows:

$$x_J^l = g\left(\sum_{I \in M} x_I^{l-1} * W_{IJ}^l + B_J^l\right) \tag{2}$$

In the equation, $g(.)$ represents the activation function, $M$ represents the set of input feature maps, $W_{IJ}^l$ represents the convolution kernel vector used between the *I*-th feature map and the *J*-th feature map. $B_J^l$ is the bias.

The cosine similarity is an index to measure the similarity between two vectors. It calculates the cosine of the angle between two vectors. The smaller the angle, the higher the correlation between the two vectors, the larger the cosine. Its value range is $(-1,1)$. Here is how the cosine similarity between $X$ and $Y$ is calculated, where $n$ is the dimension of vector.

$$\cos(X, Y) = \frac{\sum\limits_{i=1}^{n} x_i y_i}{\sqrt{\sum\limits_{i=1}^{n} x_i^2} \cdot \sqrt{\sum\limits_{i=1}^{n} y_i^2}} = \frac{\langle X, Y \rangle}{\|X\|\|Y\|} \tag{3}$$

$F_I\left(X_I^{l-1}, W_{IJ}^l\right)$ represents the similarity function between the input feature map of the *l*-th convolutional layer and the convolution kernel. $X$ represents the input feature map vector. The convolution operation based on cosine similarity can be expressed as the following equation:

$$W_{r \times z} \cdot X_{r \times z} = \sum_{i=1}^{r} \sum_{j=1}^{z} w_{ij} x_{ij} \Big/ \left(\sqrt{\sum_{i=1}^{r} \sum_{j=1}^{z} w_{ij}^2} \cdot \sqrt{\sum_{i=1}^{r} \sum_{j=1}^{z} x_{ij}^2}\right) \tag{4}$$

In Equation (4), $r \times z$ represents the size of the convolution kernel. $w_{ij}$ and $x_{ij}$ respectively represent the coefficients in the convolution kernel and the feature map. The similarity function can be expressed as:

$$F_I\left(X_I^{l-1}, W_{IJ}^l\right) = \sum_{I \in M} X_I^{l-1} \cdot W_{IJ}^l + B_J^l \tag{5}$$

Therefore, in the *l*-th convolution layer, the output based on cosine similarity operation is expressed as Equation (6)

$$\begin{aligned} x_J^l &= g(F_I) \\ W_{IJ}^l &= \left[W_{I1}^l, W_{I2}^l, \ldots, W_{In}^l\right]^T \\ X &= \left[X_1^{l-1}, X_2^{l-1}, \ldots, X_n^{l-1}\right]^T \end{aligned} \tag{6}$$

$g(.)$ represents the activation function. The higher the similarity between the input feature map and the convolution kernel $W_{IJ}^l$, the greater the output value of the convolution layer.

## 5. Results

Experimental environment: CPU Intel(R) core(7M) i7-7700ghz; Memory DDR4, 8.00G; GPU NVIDA GeForce RTX 2080 SUPER, basic frequency 1650 MHz, acceleration frequency 1815 MHz, video memory: GDDR6, 8G, 256 bit, video memory frequency 15.5 GHz, video memory bandwidth 496 GB/s.

### 5.1. Disease Area Detection and Analysis

In the training of the CNN model, Adam was used as the optimization algorithm. The learning rate was set to 0.001, the multiplier factor of learning rate decline was set to 0.1, and the cross-entropy loss function was selected as the loss function [35]. The training samples were obtained by clipping the original image, where the positive samples were the disease area and the negative samples were the non-disease. One hundred twenty-four buckwheat disease images were selected to construct an image dataset that contained only clipped images, as shown in Figure 9a,b. The positive samples were disease regions (792 regions in total), and the negative samples were non-disease regions (1135 regions in total). When data are inputted for training, the positive and negative samples are first thoroughly mixed and randomly divided in a ratio of 4:1, respectively, as training set and test set. Then, the input images were standardized by parameters with mean values of 0.471, 0.452, and 0.412, and variance of 0.282, 0.267, and 0.231, respectively.

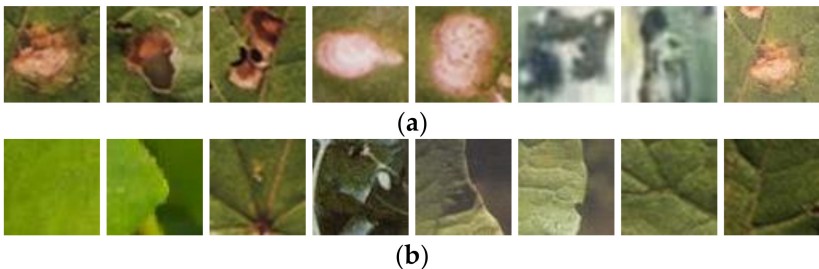

(a)

(b)

**Figure 9.** Training samples. (**a**) positive training sample; (**b**) negative training sample.

Figure 10 shows the relationship between the size of the clipping sample and the average accuracy based on CNN prediction. The experimental results show that the average accuracy of prediction tends to be stable during the 20th to 30th round of training. By comparing the different sizes of cutting samples, i.e., 24 × 16, 24 × 24, 32 × 24, 32 × 32, and 48 × 32. We find that the average prediction accuracy of 32 × 32 is higher than that of other sizes. After 20 iterations, the prediction accuracy tends to be stable. Therefore, we finally chose the size of 32 × 32 as the training data. At the same time, the candidate region obtained by this algorithm is uniformly adjusted to 32 × 32 images for classification.

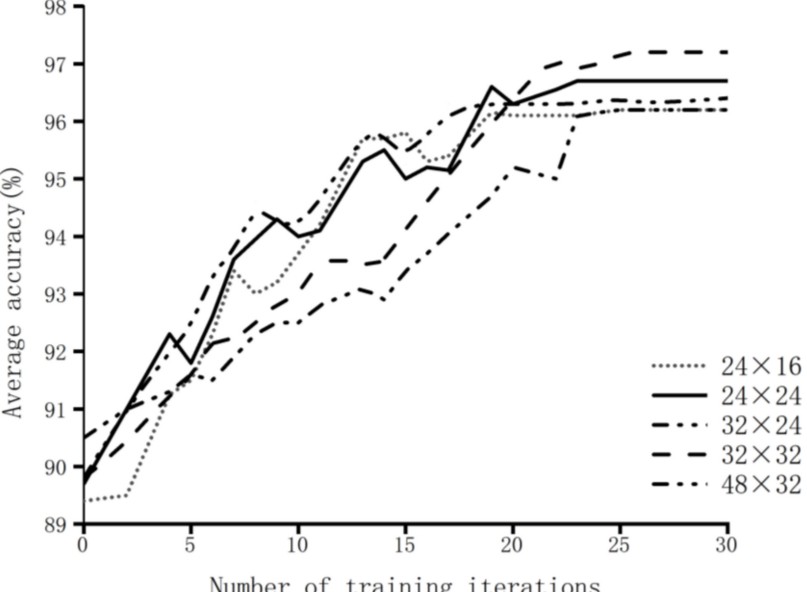

**Figure 10.** The relationship between the size of the clipping sample and the average accuracy of CNN prediction.

In this work, 500 buckwheat disease images were tested. The number of diseased spots was 16 at most and 2 at least on a single leaf. After testing and analysis, the plaques of 135 images were fully detectable. The detection rate of 117 images ranged from 100% to 90%. There were 134 images with a detection rate of less than 50 percent. As is shown in Figure 11, the horizontal coordinate represents the percentage of plaque detected in a single image, and the vertical coordinate represents the number of disease images. In general, the disease area detection rate reached 455 pictures with 70%, which had a good disease area detection effect. There are 455 images with a disease area detection rate of more than 70%, which had a good effect on disease area detection.

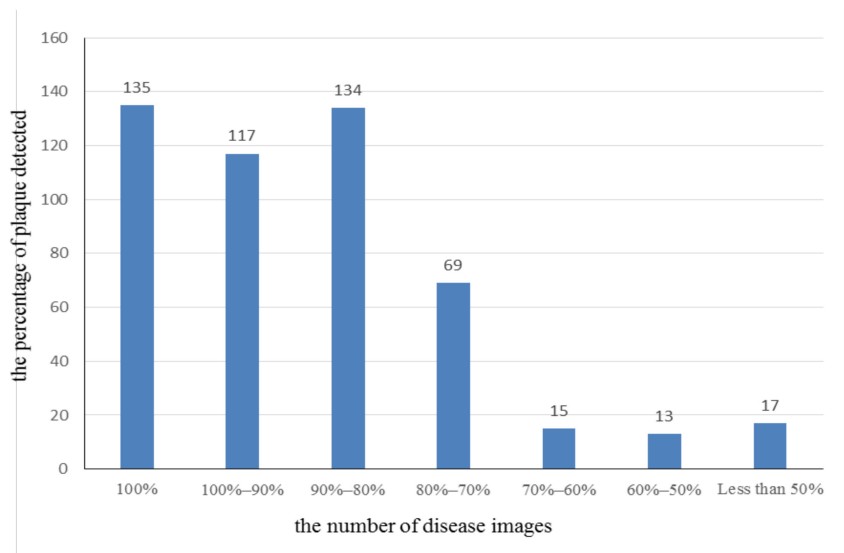

**Figure 11.** The statistics of disease area detection.

Figures 12 and 13 show the disease area obtained after CNN classification. It can be seen that the detection result is more accurate, eliminating the cross box and error detection of the disease area. Therefore, this method can accurately classify the disease area and non-disease area of buckwheat.

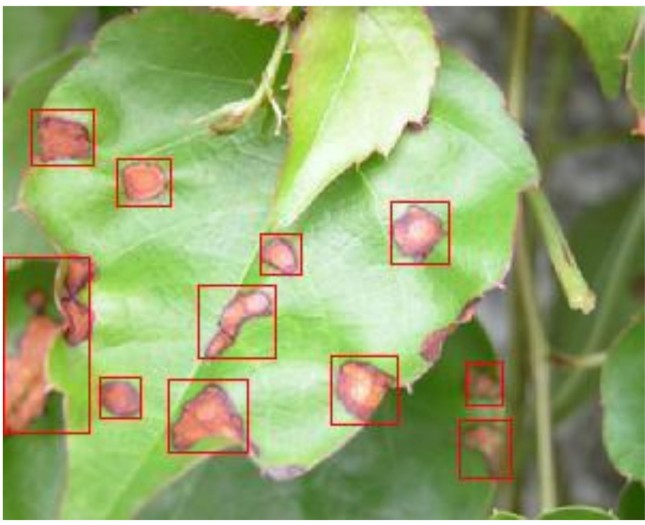

**Figure 12.** Final detection effect of buckwheat brown spot disease.

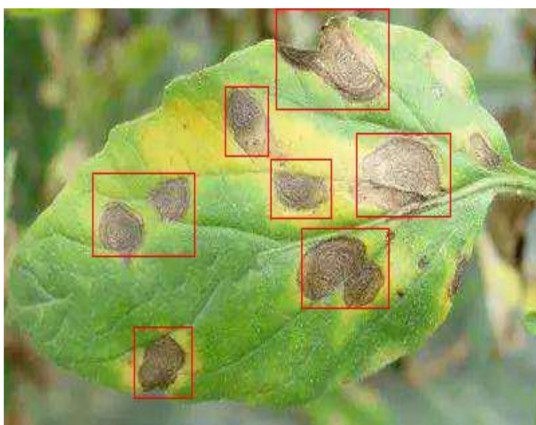

**Figure 13.** Final detection effect of buckwheat ring rot disease.

### 5.2. The Performance Analysis of Inception Module

To evaluate the performance of the inception module, 1200 samples were selected from eight types of samples, including buckwheat spot blight, buckwheat sclerotinia, buckwheat seedling blight, buckwheat ring spot, buckwheat downy mildew, buckwheat brown spot, buckwheat virus disease, buckwheat white mold, and 800 images of non-disease buckwheat were added at the same time. The training set and test set were divided into 4:1. The control variables in inception 1 and inception 2 were compared. The experimental result is shown in Table 1. Convolution layer number, convolution kernel number, learning rate, and batch size were debugged. The optimal parameters were set: 4 convolution layers, convolution kernel: (136, 200, 264, 520), learning rate: 0.02, batch size: 100. The epoch was set to 160. The experiment's results show that larger epochs could not improve performance, and further increases will result in overfitting.

**Table 1.** Performance evaluation of inception module.

| Network Structure | Iterations | Accuracy Rate (%) |
|---|---|---|
| Without the Inception | 3500 | 87.47 |
| Only the Inception 1 | 3500 | 90.28 |
| Only the Inception 2 | 3500 | 90.64 |
| The Inception 1 and Inception 2 | 3500 | **91.51** |

From Table 1, it is found that after adding the inception 1 and inception 2, the recognition accuracy of the network has been improved within the same number of iterations. After adding the Inception 1 and Inception 2 together, the accuracy rate reaches 91.51%, is higher than other structures. thus, the inception structure proposed is effective in this paper.

### 5.3. The Performance Analysis of Cosine Similarity Convolution

We compared the recognition performance of CNN based on traditional convolution, CNN based on other similarity functions, and CNN based on cosine similarity. 1200 samples were selected from eight types of samples including buckwheat spot blight, buckwheat sclerotinia, buckwheat seedling blight, buckwheat ring spot, buckwheat downy mildew, buckwheat brown spot, buckwheat virus disease, and buckwheat white mold. Then, 800 buckwheat image samples without disease were added, and the training set and test set were divided according to the ratio of 4:1. We selected the network structure determined in Section 4.2.1. For disease and non-disease identification, five experiments were performed on several convolution networks, and the results are shown in Table 2.

**Table 2.** Recognition accuracy (%) for different convolution methods.

|  | Experiment 1 | Experiment 2 | Experiment 3 | Experiment 4 | Experiment 5 | Average |
|---|---|---|---|---|---|---|
| Traditional Convolution | 93.25 | 94.85 | 94.14 | 93.48 | 94.71 | 94.07 |
| Euclidean Distance Convolution | 92.31 | 92.67 | 92.71 | 92.19 | 92.49 | 92.47 |
| Chebyshev Distance Convolution | 94.18 | 94.58 | 93.82 | 94.17 | 94.38 | 94.23 |
| Manhattan Distance Convolution | 92.67 | 92.13 | 93.28 | 92.35 | 93.16 | 92.72 |
| Cosine Similarity Convolution | **98.13** | **98.28** | **98.42** | **98.63** | **98.38** | **98.37** |

Compared with the traditional convolution method, it can be observed from Table 2 that the recognition accuracy of the convolution based on cosine similarity is improved, and the average accuracy rate increased by 4.14%. However, the convolution network based on other similarity functions has lower accuracy than the traditional convolution network. This shows that the calculation methods of other similarity functions amplify the differences between the features in the sample, making the sample noise greatly interfere with the process of feature extraction. The cosine similarity will limit the output result between −1 and 1, which can minimize the impact of noise on feature extraction.

In order to further analyze the performance of the cosine similarity convolutional network, there are three experiments; we give the loss function and accuracy curve for the traditional convolution network and the cosine similarity convolution network, as shown in Figures 14 and 15.

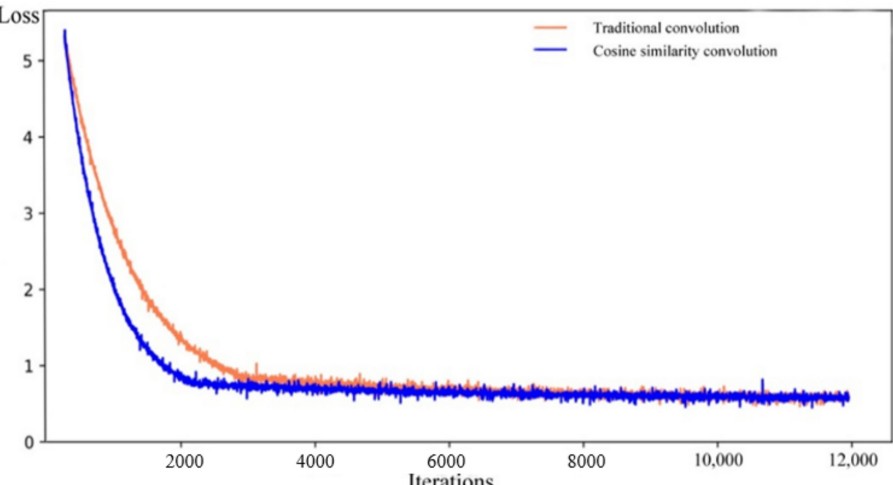

**Figure 14.** Loss curve for training.

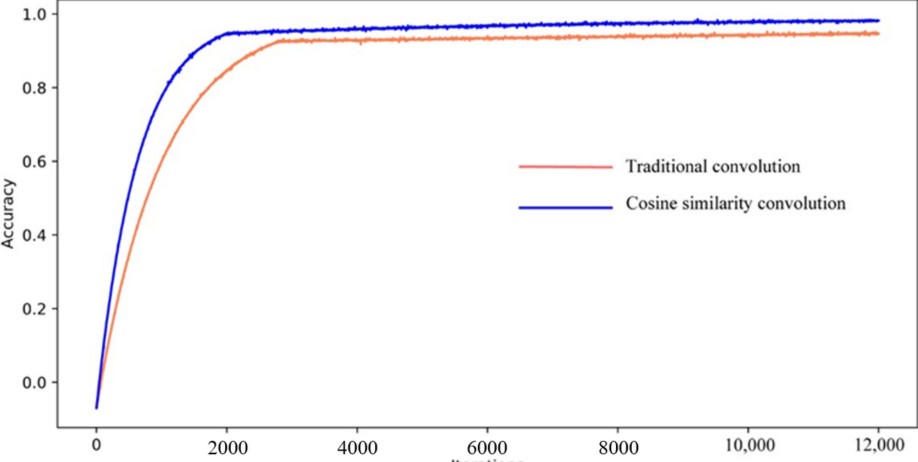

**Figure 15.** Accuracy curve for training.

It can be seen from Figures 14 and 15 that the network of traditional convolution gradually converges after about 6000 iterations, while the convolution based on cosine similarity converges after 4000 iterations. Therefore, the network based on cosine similarity convolution converges faster and finds the global optimal solution more easily.

The convolution method based on cosine similarity can better evaluate relevancy between the convolution kernel and the features of the input feature map. In this way, a larger activation value can be obtained at a location similar to the convolution kernel feature, while avoiding the influence of noise on feature extraction. It can be found from our study that the convolution based on cosine similarity is more accurate than other convolution in the characterization of buckwheat disease. It is insensitive to noise, and its recognition accuracy is higher than other convolution methods. At the same time, the calculation of cosine similarity convolution is fast, iteration steps are fewer, so the network has a shorter operation time.

Due to the influence of sampling period and environment, there are many samples with uneven illumination in the image of buckwheat disease. In this case, the gray value of the image changes significantly, so the activation value obtained by the traditional convolution operation will also change suddenly. Figure 16 shows the output of traditional convolution and cosine similarity convolution.

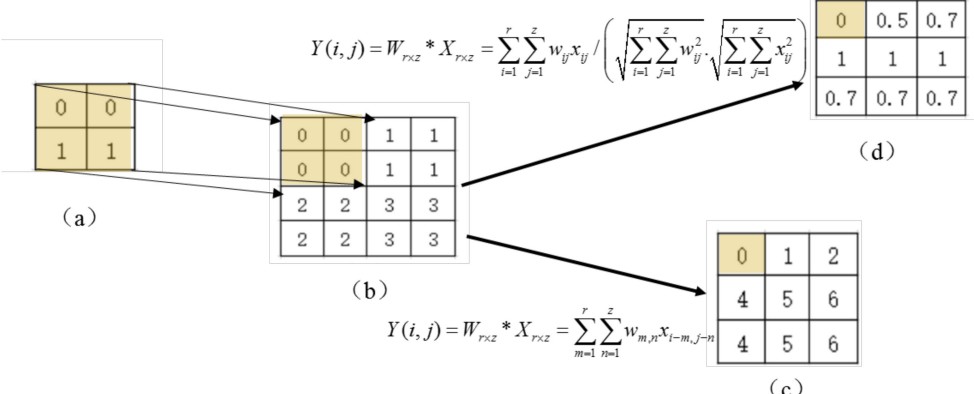

**Figure 16.** Comparison of two convolution operations. (**a**) Convolutional kernel; (**b**) Input feature map; (**c**) Traditional convolution output; (**d**) Convolution output based on cosine similarity.

Assuming that the values in Figure 16 represent the pixel gray values of the image, and the difference in these values is mainly caused by the uneven illumination, the convolution kernel in Figure 16a and the input feature map in Figure 16b are used for traditional convolution and cosine similarity based convolution respectively. It can be seen that the output value obtained by the traditional convolution method has obvious differences, and its feature extraction ability is correspondingly weakened, which is obviously not the result we expected. The output value based on the cosine similarity convolution was uniform, which shows that the method can better adapt to the uneven illumination and is more conducive to feature extraction.

Figure 17 shows the output feature maps of buckwheat leaf under different lighting environments after implementing the convolution operation based on cosine similarity. In the first layer of the convolution feature map, besides the sample contour, there are some obvious boundaries caused by uneven illumination; After the second convolution operation, the feature map is weak to uneven illumination; In the third convolution layer, the feature map basically eliminates the impact of illumination; the fourth convolution layer and the fifth convolution layer, all feature maps tend to retain only the high-level feature of buckwheat leaves. Therefore, it can be concluded that the hue of the feature image is relatively balanced, which is obtained by convolution based on cosine similarity. It indicates that this method can reduce the impact of uneven illumination and has better feature extraction capabilities.

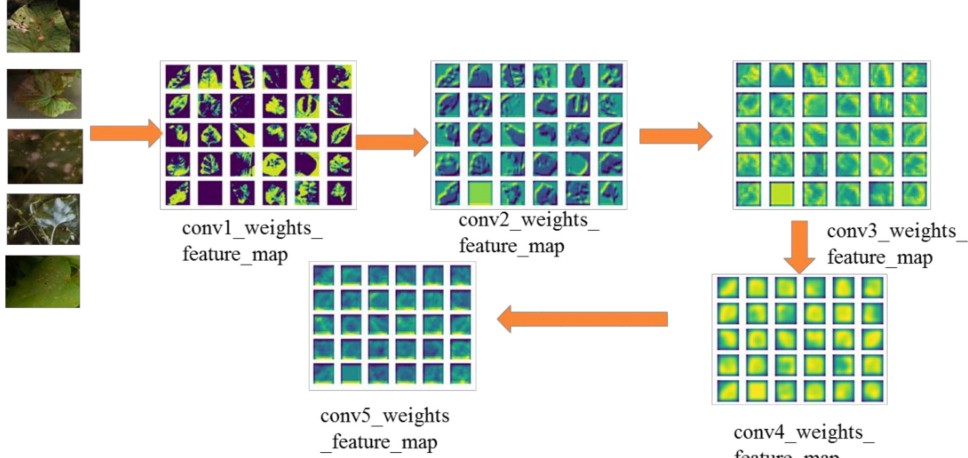

**Figure 17.** Convolution feature map of buckwheat leaf under uneven illumination.

### 5.4. The Recognition of Buckwheat Diseases

The inception structure of Section 4.2.1 and the CNN with cosine similarity convolution were adopted to recognize buckwheat disease. We compared the final recognition accuracy and used Precision, Recall, and F1-measure to evaluate the recognition effect [36].

The experimental results are shown in Table 3, in addition to the comparison with mainstream CNN models (AlexNet, VGG, GoogleNet, ResNet, LeNet, Faster R-CNN, R-FCN, FPN, YOLOv3, ZFNet) [36], in order to demonstrate the performance of the proposed method, we also introduced the face recognition model, which is currently well studied to test in data set of the buckwheat disease. The results show that accuracy, precision, recall, and F1-measure reached 96.43, 96.82, 95.62, and 96.71%, respectively, for our method. Compared with AlexNet, VGG, GoogleNet, ResNet and Faster R-CNN, R-FCN, YOLOv3 LeNet, ZFNet, the optimal performance of our method was improved by 1.47, 2.1, 1.17, and 3.03%, respectively. The FPS (average number of pictures processed per second) of our method is 5.19, which shows that its processing speed is also at a high level. We selected some models to draw ROC curves on validation sets and test sets, as follows in Figure 18. It can be seen that ROC curves on the test set and validation set are basically consistent; therefore, our model is stable and performance is optimal.

**Table 3.** Results of buckwheat disease identification.

| Model | Accuracy (%) | Precision (%) | Recall (%) | F1-Measure (%) | AUC (%) | FPS (s) |
|---|---|---|---|---|---|---|
| AlexNet | 87.31 | 84.92 | 88.57 | 90.34 | 96.13 | 3.15 |
| Vgg-16 | 89.27 | 88.42 | 90.35 | 89.53 | 94.12 | 3.18 |
| GoogleNet | 90.53 | 92.84 | 91.47 | 90.53 | 95.35 | 4.15 |
| ResNet | 94.75 | 93.72 | 92.37 | 93.41 | 97.18 | 4.17 |
| Faster R-CNN | 93.24 | 92.34 | 93.72. | 91.58 | 95.36 | 5.37 |
| R-FCN | 94.22 | 94.31 | 94.36 | 92.15 | 95.31 | 4.85 |
| FPN | 94.16 | 94.83 | 92.51 | 93.87 | 97.78 | 4.27 |
| YOLOv3 | 95.03 | 94.72 | 94.51 | 92.67 | 96.46 | 4.53 |
| LeNet | 94.57 | 94.23 | 92.59 | 91.88 | 95.12 | 5.21 |
| ZFNet | 94.21 | 94.17 | 93.23 | 93.42 | 97.43 | 5.33 |
| **Ours (inception +Cosine similarity convolution)** | **96.43** | **96.82** | **95.62** | **96.71** | **98.21** | **5.19** |
| DeepFace | 91.53 | 90.31 | 90.73 | 89.35 | 94.37 | 5.18 |
| VGGFace | 91.82 | 91.24 | 89.42 | 90.17 | 95.52 | 5.16 |
| FaceNet | 90.47 | 89.27 | 90.33 | 90.16 | 96.29 | 4.21 |
| DeepID2+ | 93.82 | 93.57 | 92.49 | 90.38 | 96.17 | 4.17 |
| WST Fusion | 92.73 | 92.47 | 93.21 | 92.54 | 97.05 | 3.18 |
| SphereFace | 93.68 | 94.38 | 92.86 | 92.67 | 97.46 | 3.23 |
| RangeLoss | 91.79 | 92.45 | 91.65 | 90.78 | 95.37 | 3.21 |
| HiReST-9+ | 88.36 | 88.67 | 89.59 | 89.76 | 94.88 | 5.79 |

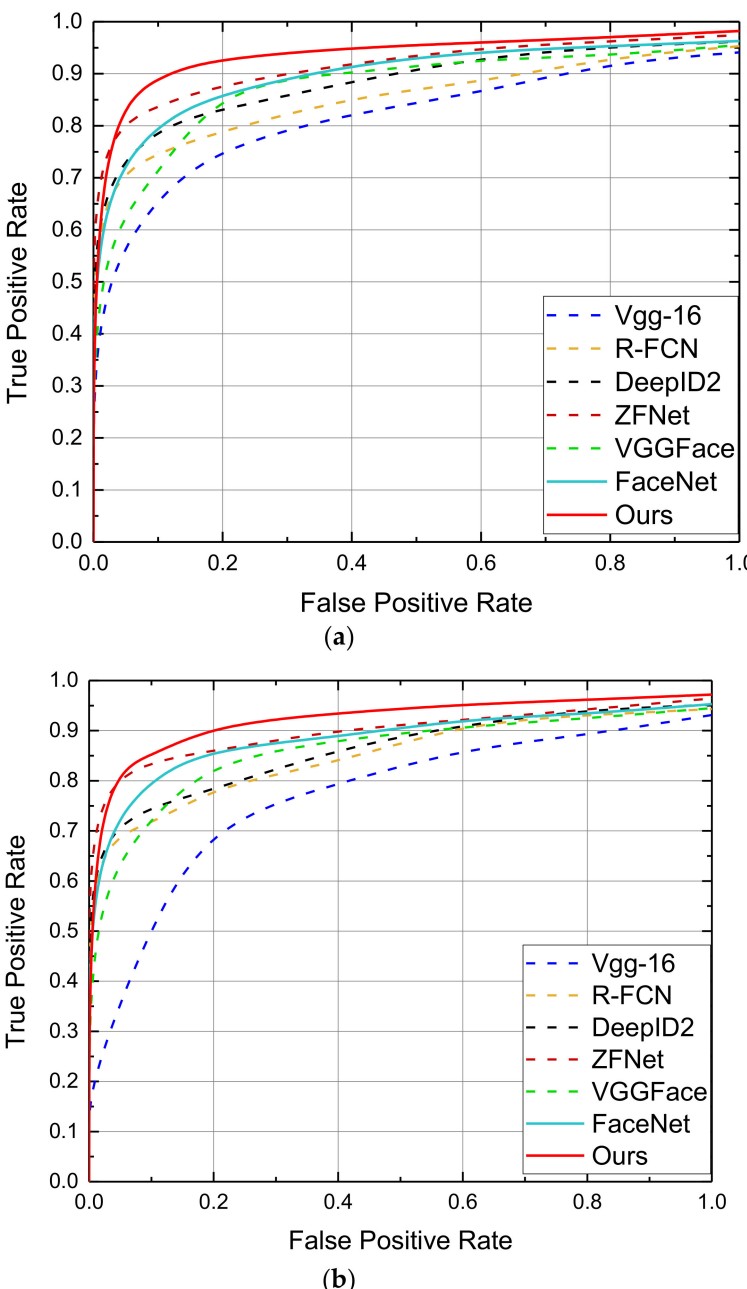

**Figure 18.** ROC curve for different models on validation sets and test sets. (**a**) ROC curve for different models on testing sets; (**b**) ROC curve for different models on validation sets.

In order to further study the model effect for buckwheat disease recognition after the disease areas of buckwheat leaf are detected, we tested whether to perform area detection. After the method of disease area detection was added in Section 4.1, the recognition effect was significantly improved, as shown in Table 4. After the method of disease area detection was adopted, the accuracy, precision, recall, and F1-measure of our method reached 98.13, 97.54, 96.38, and 97.82%, respectively, which was an increase of 1.7, 0.72, 1.76, and 1.11% compared without disease area detection. Combined with the analysis of Table 3 it is found that after the disease area detection is adopted, the average accuracy precision, recall, and F1-measure of each recognition model increased by 1.3, 1.51, 1.69, and 1.48%, respectively. However, we found that area detection introduces additional computational overhead, so the processing speed is slightly reduced. The PFS of all models decreased by an average of 0.46, but it has little effect on the overall performance. To verify the advantages of our structure, the cosine similarity convolution was added to ResNet, Faster R-CNN, R-FCN,

FPN, YOLOv3, LeNet, and ZFNet. The results are shown in Figure 19. It can be seen that the structure presented in this paper has a better effect on the discrimination of buckwheat diseases compared with other networks. ROC curves on validation sets and test sets, as shown in Figure 20. The model performance is similar in the validation set and test set. In fact, the effect on the test is a little more than the effect on the validation set.

**Table 4.** Identification results of Buckwheat diseases after region detection.

| Model | Accuracy (%) | Precision (%) | Recall (%) | F1-Measure (%) | AUC (%) | FPS (s) |
|---|---|---|---|---|---|---|
| AlexNet | 87.93 | 86.37 | 88.92 | 92.34 | 97.12 | 3.12 |
| Vgg-16 | 90.17 | 90.17 | 91.83 | 90.27 | 95.73 | 2.87 |
| GoogleNet | 93.42 | 94.21 | 93.84 | 92.67 | 96.59 | 3.87 |
| ResNet | 93.78 | 94.92 | 94.42 | 94.89 | 97.37 | 3.72 |
| Faster R-CNN | 95.16 | 95.25 | 94.57 | 92.54 | 96.86 | 3.81 |
| R-FCN | 95.26 | 96.13 | 95.38 | 92.87 | 97.12 | 3.85 |
| FPN | 96.37 | 96.21 | 94.36 | 95.17 | 97.84 | 3.57 |
| YOLOv3 | 95.57 | 95.32 | 96.83 | 95.37 | 98.03 | 4.12 |
| LeNet | 94.87 | 95.17 | 93.27 | 92.68 | 95.68 | 4.27 |
| ZFNet | 95.78 | 94.35 | 95.12 | 93.71 | 96.73 | 4.03 |
| **Ours (inception + Cosine similarity convolution)** | **98.13** | **97.54** | **97.38** | **97.82** | **98.89** | **4.31** |
| DeepFace | 94.73 | 93.28 | 92.67 | 90.48 | 95.42 | 4.43 |
| VGGFace | 93.02 | 92.83 | 91.61 | 92.35 | 96.74 | 4.32 |
| FaceNet | 92.57 | 91.67 | 92.57 | 92.67 | 96.81 | 3.49 |
| DeepID2+ | 94.24 | 94.38 | 93.87 | 92.49 | 97.13 | 3.53 |
| WST Fusion | 93.86 | 94.35 | 95.61 | 93.49 | 97.41 | 3.02 |
| SphereFace | 94.56 | 94.85 | 93.72 | 93.57 | 97.58 | 3.13 |
| RangeLoss | 92.71 | 93.73 | 92.71 | 92.18 | 96.93 | 2.97 |
| HiReST-9+ | 90.73 | 90.32 | 91.49 | 91.33 | 95.93 | 4.27 |

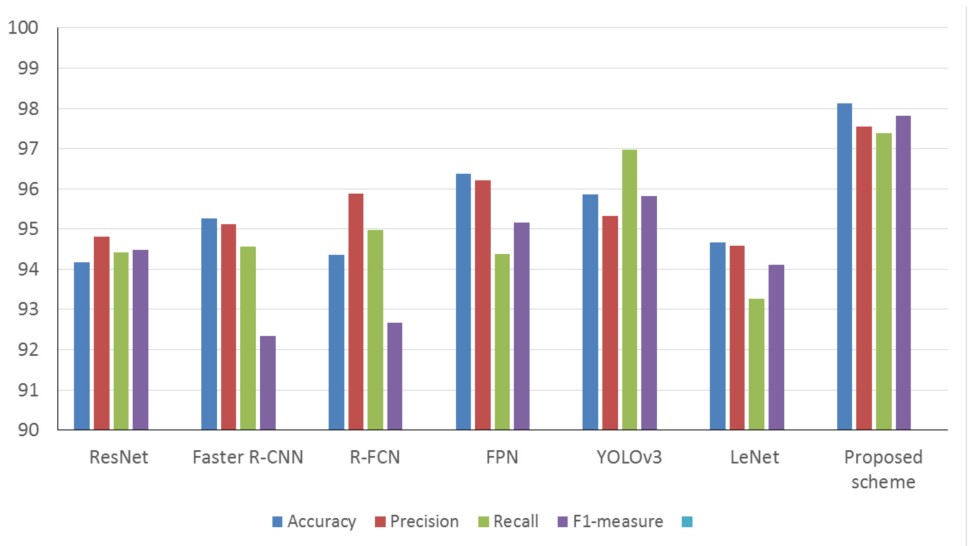

**Figure 19.** The cosine similarity convolution was added to other networks.

We also classified buckwheat spot blight, buckwheat sclerotinia, buckwheat seedling blight, buckwheat ring spot, buckwheat downy mildew, buckwheat brown spot, buckwheat virus disease, and buckwheat white mold. Table 5 shows the results of specific disease classification using the identification framework proposed in this article after the disease area detection. It can be seen that the recognition effect for specific diseases has been significantly reduced, especially for buckwheat downy mildew. Because it is a binary classification problem to discriminate whether there are diseases in buckwheat, it is simply, the classification effect is better. If we want to identify specific disease types, the difficulty of identification will increase as the number of classification targets increases. Due to the different performance of the photography equipment and different photography environments, the imaging quality is quite different; Moreover, the accuracy of recognition largely

depends on the training of the model and the number of samples. The samples in this article comes from field collection and is limited by conditions. There are only 500 samples of each disease, and the model training is insufficient, so the recognition effect is greatly affected. However, it can be seen that the accuracy, precision, recall, and F1-measure of buckwheat spot blight and buckwheat ring disease still reach more than 90%. This is because the edge contours of these two diseases on buckwheat leaves are clear. The feature map of the disease can be obtained accurately when the convolutional neural network is used for feature extraction, so it shows higher accuracy in classification. In order to analyze the recognition accuracy of our method in detail, the confusion matrix is used to represent the classification results. In Figure 21a, the diagonal represents the number of correct classifications for each classification; In Figure 21b, the diagonal represents the recognition accuracy of each classification. Although there is a difference in the recognition accuracy, on the whole, the recognition effect of spot disease is better than that of mycosis. The regional feature of leaf spot disease are obvious for buckwheat, which has high recognition, so regional features can be accurately extracted. However, the distribution of mycosis on buckwheat leaves is scattered and the edge is fuzzy, it is difficult to extract the feature.

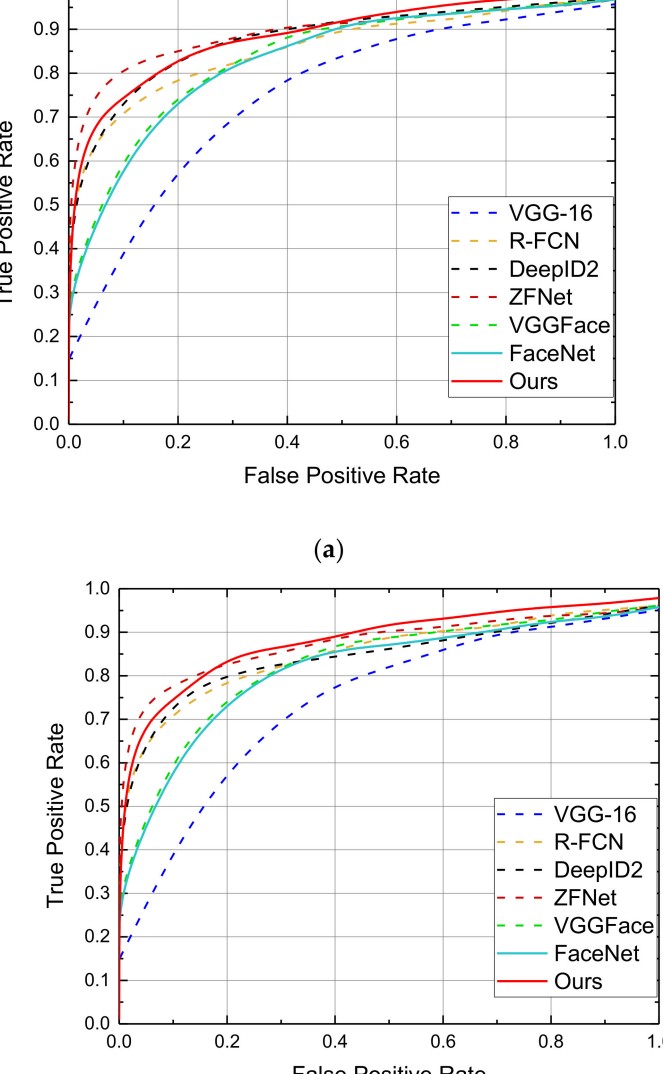

(**a**)

(**b**)

**Figure 20.** ROC curve for different models on validation sets and test sets after region detection. (**a**) ROC curve for different models on test sets; (**b**) ROC curve for different models on validation sets.

**Table 5.** Effect of different buckwheat diseases for their identification.

| Disease Type | Accuracy (%) | Precision (%) | Recall (%) | F1-Measure (%) | AUC (%) |
|---|---|---|---|---|---|
| Spot blight | 90.37 | 90.51 | 91.35 | 92.92 | 96.78 |
| Sclerotinia | 83.53 | 82.48 | 83.41 | 80.54 | 87.32 |
| Seedling blight | 79.18 | 78.38 | 80.72 | 80.21 | 86.75 |
| Ring spot | 91.43 | 91.57 | 92.61 | 92.39 | 96.41 |
| Downy mildew | 78.31 | 78.45 | 75.28 | 77.38 | 82.13 |
| Brown spot | 86.93 | 85.91 | 88.47 | 86.52 | 89.57 |
| Virus disease | 87.42 | 86.19 | 87.43 | 86.52 | 88.46 |
| White mold | 85.48 | 85.36 | 86.92 | 86.74 | 87.59 |

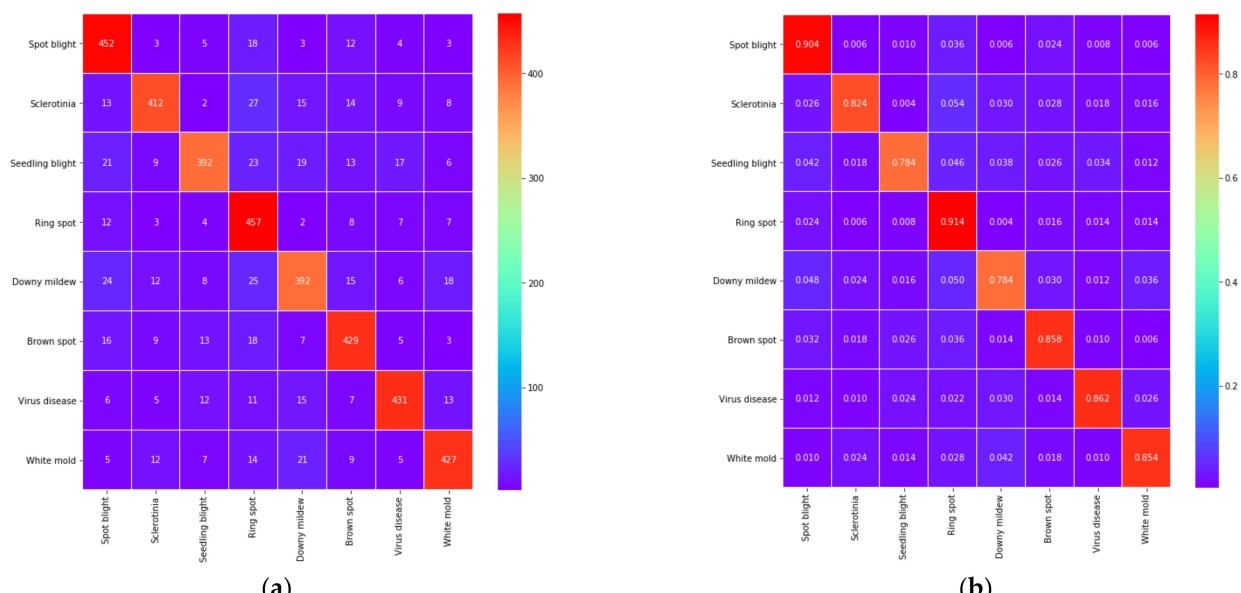

**Figure 21.** Buckwheat disease classification; (**a**) confusion matrix, (**b**) normalized confusion matrix.

## 6. Conclusions and Future Directions

In the article, the main contribution is to realize accurate disease area detection by MSER and CNN, and a two-level inception structure was added to improve classification accuracy. Furthermore, in order to eliminate the interference in disease identification, which is caused by illumination imbalance, cosine similarity is introduced in the convolution process. In this work, the CNN structure based on inception combined with cosine similarity was better than the current recognition framework in the performance for buckwheat diseases. Compared with recognition frameworks, such as the AlexNet, VGG, GoogleNet, ResNet et al., the accuracy, precision, recall, and F1-measure, our method got a consistent outperformance regarding precision, recall, and F1-score. In particular, the proposed method is robust when the light is not uniform and the leaves are crossed and overlapped. Because our method detects the disease area, it results in a small increase in processing time, but overall performance is not affected. Due to the similar symptoms of buckwheat downy mildew and buckwheat Seedling blight, the accuracy of recognition is not high. In future plans, we will further improve the identification accuracy of these two diseases.

**Author Contributions:** X.L. completed theoretical analysis and model design; S.Z. completed the data analysis; S.C. carried out implementation and data tests on the model; Z.Y. provides basic data and disease identification; R.Y. completed the program; H.P. sorted out the papers and completed the drawings. All authors have read and agreed to the published version of the manuscript.

**Funding:** This work was supported by NSFC Grant Nos. 61701060 and 61801067, Guangxi Colleges and Universities Key Laboratory of Intelligent Processing of Computer Images and Graphics Project No. GIIP1806, and the Science and Technology Research Project of Higher Education of Hebei

Province (Grant No. QN2019069), and Chongqing Key Lab of Computer Network and Communication Technology (CY-CNCL-2017-02).

**Institutional Review Board Statement:** Not applicable.

**Informed Consent Statement:** Not applicable.

**Data Availability Statement:** Not applicable.

**Acknowledgments:** We would like to acknowledge the support for our work from the researchers of Chongqing Key Lab of Computer Network and Communication Technology.

**Conflicts of Interest:** There are no conflicts of interest in this work.

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
