# Peer review of "Buckwheat Disease Recognition Based on Convolution Neural Network"

_applsci, doi:10.3390/app12094795_

Round 1

Reviewer 1 Report

The article is devoted to the applied solution of the problem of classification by artificial intelligence methods. The topic of the article is relevant. The structure of the article differs from the classic one for technical research articles. The level of English is acceptable. The article is easy to read. Tables and figures of acceptable quality. References cites 35 current sources.

The following recommendations can be formulated for the material of the article:

  1. First of all, I pay tribute to the courage of the authors. Writing articles about applications of convolutional neural networks is very difficult. A lot has already been written. It is necessary to formulate the scientific novelty very clearly and very strictly prove its adequacy. All the more strange is the 9-line Discussion section. Do the authors have anything to say? Where did the Conclusions go? Where is the main contribution formulated?
  2. The authors put a "muzzle" on the convolutional network, which implements the operation of calculating the cosine distance. It's good. True, this technique, like the distance of Karunen Loev, Bhattacharya, Renier and other methods of factor analysis, is more appropriate to apply to an independently organized factor space. This makes it easier to control the result. How did the authors determine the edge when the dimension of the output feature space becomes so small that it begins to negatively affect the recognition result? Are the authors sure that the material proposed in section 2.4.2 is new?
  3. Metric (7)-(10) depends on the decision threshold. How did the authors take this into account?
  4. The authors compared their model with AlexNet, VGG, GoogleNet, ResNet, LeNet, Faster R-CNN, R-FCN, FPN, YOLO v.3. It's impressive. And why did they forget ResNet and ZFNet?

Reviewer 2 Report

The paper applies CNN to classify and detect Buckwheat diseases. The first point is the paper sequence and sections. It should be re-organized. Section 3 Results appears twice. There is no section 2.3. 
My suggestion is section 2 background about the CNN (current section 2.2), Inception layer and normal convolution, and cosine convolution.
Section 3 describes the dataset.
Section 4 the paper contributions: MSER detection, inception layer, and cosine convolution as well as the new CNN architecture proposed in the paper.
Section 5 to show the results: 5.1 Mser detection and Alexnet (current pages 7,8,9)
5.2 table 1 experiment and inception, 5.3 cosine convolution results, and table 2 and figures...
5.4 CNN comparisons
There are important points to clarify. 
First, Alexnet is applied to the entire image to classify the disease occurrence, or does it considers only the MSERs regions?
The authors claim "Pooling process: Pixels in the neighborhood are summed to get a pixel,"? Please, clarify if it is not max pool?
If the Alexnet input is 128x128, please clarify what is the different sizes of cutting samples?
the inception uses 64x64 inputs, all networks use the same input size?
What is the difference in parameters number with and without inception layers, with and without the cosine convolutions? Number of operations (mul and add)
It is stranger in table 3 compares CNN and object detection CNN (yolo, faster R-cnn). 
Even lenet, a simple network that achieves good results? 
please, add the input size and the number of layers, and the total of parameters in table 3 e 4.

Please, the dataset and the experiment may be shared? to validate the results?

Reviewer 3 Report

Dear authors,

Please find my various remarks in the attached PDF file. 

Kind regards,

Reviewer 4 Report

I have the following recommendations regarding improvemnt of the paper.

1. It would be better if the paragraph in Lines#100-130, is split to two paragraphs.
2. Before using an abbreviation, the authors should explain the acronym i.e. LSTM in line#126 (an example). Please check all the acronyms you have used throughout the paper. 
3. Keep a space between sentences i.e. check on line#117. Please check throughout the paper for these kind of mistakes.
4. Figure 1 if split int Figure 1. (a), (b), (c),...can increase the readibility and interest of the readers.
5. Redraw figure 7, with proper explanation of the legends.
6. Figure 12 and Figure 13 are separate figures, hence keep figure 13 below the figure 12. if you denote it as (a), and (b) subplots, then you can number it as Figure 12, with (a), (b) subplots. Authors are advised to properly place these figures.
7. In figure 18, instead of "Ours", the authors can write the scheme or the "Proposed scheme".
8. Section 4 is named as "Discussion". I think this discuss the conclusion and future directions. Hence appropriate title can be "Conclusion and Future Directions".
9. At the end of the Introduction section, write one paragraph that discuss about the contents of each section i.e. Section 1 is about ..., In section 2 we discuss...., Section 3 illustrate....etc.
10. Authors should mention a section after related works i.e. the research gap and contributions of the paper.

Round 2

Reviewer 1 Report

My comments on the first version of the article were:

  1. First of all, I pay tribute to the courage of the authors. Writing articles about applications of convolutional neural networks is very difficult. A lot has already been written. It is necessary to formulate the scientific novelty very clearly and very strictly prove its adequacy. All the more strange is the 9-line Discussion section. Do the authors have anything to say? Where did the Conclusions go? Where is the main contribution formulated?
  2. The authors put a "muzzle" on the convolutional network, which implements the operation of calculating the cosine distance. It's good. True, this technique, like the distance of Karunen Loev, Bhattacharya, Renier and other methods of factor analysis, is more appropriate to apply to an independently organized factor space. This makes it easier to control the result. How did the authors determine the edge when the dimension of the output feature space becomes so small that it begins to negatively affect the recognition result? Are the authors sure that the material proposed in section 2.4.2 is new?
  3. Metric (7)-(10) depends on the decision threshold. How did the authors take this into account?
  4. The authors compared their model with AlexNet, VGG, GoogleNet, ResNet, LeNet, Faster R-CNN, R-FCN, FPN, YOLO v.3. It's impressive. And why did they forget ResNet and ZFNet?

Of the four remarks, the authors considered three and a half worthy of an answer. They agreed with the first remark. On the second remark, I received the answer - "CNN knows best", on the third - "Everyone does it this way", on the fourth - well done. I especially didn't like the answer to the third remark. It is too early to ignore the decision threshold in evaluating classification results than to measure speed in centimeters. It seems that they measured correctly, but without measuring time, the result is unscientific. I recommend the authors to read about DET-curves.

Reviewer 2 Report

The revised paper has a coehrent structure. My questions have been answered.

Author Response

Thanks reviewer for the valuable comment. 

Reviewer 3 Report

I would like to thank the authors for the various modifications. 

Here are my new comments:

1°) I think it is still important to distinguish the results obtained on both the training and the test sets.

I think that the results should be displayed to show the robustness of the method and to highlight its generalization. In addition, it is stated that the results are separated with a 4:1 ratio for the training and test sets, but not that the actual results are from the training or test sets.

Generally and methodically to learn a model and avoid overfitting, the database is divided into 3 sets: training (70%), validation (10%) and test (20%), percentages are in example.

During the learning process, the training and validation curves are displayed (here there is only the training loss curve). The validation set allows to check that there is no overfitting during the training of the model. Finally, the last phase which really validates the model is the evaluation of the model performances on the test base. 

2°) As mentioned in your report, the number of epochs of 160, should also be mentioned in the manuscript. I find this number of epochs very consequent, hence the importance of proving that there is no overfitting.

3°) Precision and Recall are often in tension. The overall performance of a classifier, summarized over all possible thresholds, is given by the Receiver Operating Characteristics (ROC) curve. In that way, I don't think all the metrics are still useful in Table 4.

4°) Many mathematical notations are still displayed as exponents, they should be corrected.

Some typo errors:

  • Line 130: space is missing between "LSTM" and (...)
  • Line 439: spaces missing in the parenthesis
  • Line 525: space missing

With the consideration of the above remarks, I am for acceptance of the paper.
